# Towards Certified Probabilistic Robustness with High Accuracy

## Abstract

Adversarial examples pose a security threat to many critical systems built on neural networks (such as face recognition systems and self-driving cars). While many methods have been proposed to build robust models, how to build certifiably robust yet accurate neural network models remains an open problem. For example, adversarial training improves empirical robustness, but they do not provide certification of the model's robustness. Conversely, certified training provides certified robustness but at the cost of a significant accuracy drop. In this work, we propose a novel approach that aims to achieve both high accuracy and certified probabilistic robustness. Our method has two parts which together achieve our goal, *i.e.*, a probabilistic robust training method with an additional goal of minimizing variance in divergence in a given vicinity and a runtime inference method for certified probabilistic robustness of the predictions. Compared to alternative methods such as randomized smoothing and certified training, our approach avoids introducing strong noise during training, is effective against a variety of perturbations, and most importantly, achieves certified probabilistic robustness without sacrificing accuracy. Our experiments on multiple models trained on different datasets demonstrate that our approach significantly outperforms existing approaches in terms of both certification rate and accuracy.

## 1 Introduction

Neural networks are increasingly adopted in many domains, including security-critical systems such as self-driving cars (Kurakin et al., 2017b) and face-recognition-based authentication systems (Sharif et al., 2016). Meanwhile, various safety and security issues of neural networks are identified as well. Arguably the most notable one is the presence of adversarial examples. Adversarial examples are inputs that are carefully crafted by adding human imperceptible perturbation to normal inputs to trigger wrong predictions (Kurakin et al., 2017a). Their existence poses a significant threat when the neural networks are deployed in security-critical scenarios. For example, adversarial examples can mislead road sign recognition systems of self-driving cars and cause accidents (Kurakin et al., 2017b). In other use cases, adversarial examples may allow unauthorized access through face-recognition-based authentication (Sharif et al., 2016).

To defend against adversarial examples, various methods for improving a model's robustness have been proposed. Two prominent categories are adversarial training (Bai et al., 2021; Wong et al., 2020) and certified training (Müller et al., 2022; Shi et al., 2021), both of which aim to improve a model's accuracy in the presence of adversarial examples whilst maintaining their accuracy with normal inputs if possible. Adversarial training works by training the neural network with a mixture of normal and adversarial examples. The latter may be either generated before hand (Miyato et al., 2019) or during the training (*e.g.*, min-max training (Zhang et al., 2019a)). While empirical studies show that adversarial training often improves a model's robustness whilst maintaining model accuracy, it does not offer any formal guarantee of model robustness (Zhang et al., 2019b), rendering it less than ideal. For instance, a model trained through adversarial training can still be vulnerable to new threats such as adaptive adversarial attacks (Liu et al., 2019a; Tramer et al., 2020).

Certified training aims to provide a certain guarantee of robustness. These methods typically incorporate robustness verification techniques (Xu et al., 2020) during training, *i.e.*, they aim to find a valuation of network parameters such that the model is provably robust with respect to the training

samples. While they may certify the model robustness on some input samples, they often reduce the model's accuracy significantly (Chiang et al., 2020). Recent studies have shown that state-of-the-art certified training can result in up to 70% accuracy drop on MNIST and 90% on CIFAR-10 (Chiang et al., 2020). This is unacceptable for many real-world applications. Furthermore, due to the complexity of neural network verification, such techniques are often limited to small or medium models and limited kinds of perturbations (Müller et al., 2022). Therefore, there is a pressing need for an effective and efficient approach that can achieve both high accuracy and certified robustness. An alternative method to certified training is randomized smoothing (Cohen et al., 2019) which certifies certain forms of robustness (*e.g.*, robustness within some $L^2$-norm) by systematically introducing strong noises during training. It however suffers from the same problem of significant accuracy loss.

In this work, we introduce a method that certifies a model's *probabilistic* robustness whilst maintaining its accuracy. Our method is designed based on the belief that deterministic robustness (*i.e.*, a model always makes the same decision within a certain vicinity) is often infeasible without seriously compromising accuracy, whereas probabilistic robustness (*e.g.*, a model makes the same prediction most of the time within a certain vicinity) is often sufficient in practice. Our approach comprises two parts, *i.e.*, a novel probabilistic robust training method that minimizes divergence variance, and a runtime inference method to certify the model's probabilistic robustness. In the training phase, our approach focuses on minimizing variance across model predictions on similar inputs to improve the robustness. Unlike existing adversarial training methods that focus on one specific group of adversarial attacks, *e.g.*, PGD-based adversarial training (Zhang et al., 2019a) relies on the PGD attack (Madry et al., 2018), our method improves the model's robustness without overfitting to specific adversarial attacks. Furthermore, our approach can be easily applied to handle a variety of different perturbations, such as rotation and scaling. Note that unlike randomized smoothing, our method does not introduce noise during training. In the inference phase, our approach certifies the model's probabilistic robustness by considering a given input in its peripheral region. We show that the probabilistic certified robustness of a model can be derived from the accuracy of the model in the peripheral region.

We evaluate our method by training models on multiple standard benchmark datasets and compare them with state-of-the-art robustness-improving methods, including adversarial training, certified training and others. We compare our approach with eight baseline approaches in terms of standard accuracy (*i.e.*, accuracy on normal test data), adversarial accuracy (*i.e.*, accuracy in the presence of adversarial attacks), certified robustness rate (*i.e.*, the probability of a test sample on which the model's probabilistic robustness is successfully certified), and certified robust accuracy (*i.e.*, probability of a test sample being certified robust and correct). Compared to the state-of-the-art adversarial training, we show that our method achieves a competitive or higher adversarial accuracy while sacrificing significantly less standard accuracy (*i.e.*, up to 50% less). More importantly, we are able to certify the model's robustness with regards to most of the test inputs (*i.e.*, up to 96.8% on MNIST and 92% on CIFAR-10). Compared to the state-of-the-art certified training, our method achieves a highly robust model whilst maintaining the model's accuracy, *i.e.*, our standard accuracy is almost twice as high as that of certified training. Overall, the experiments show our method achieves a high level of certified robustness whilst maintaining the model accuracy.

## 2    BACKGROUND AND PROBLEM DEFINITION

In standard supervised learning, a neural network model is a function that takes inputs from $\mathcal{X}$ and produces outputs in $\mathcal{Y}$, where $\mathcal{X}, \mathcal{Y}$ are sets of inputs and outputs, respectively. Suppose we have a hypothetical function $\bar{h} : \mathcal{X} \to \mathcal{Y}$ that we want to approximate using a neural network model given as $h : \mathcal{X} \to \mathcal{Y}$. For any input $x$ in $\mathcal{X}$, the neural network model $h$ produces a prediction $h(x)$.

With ground-truth label $\bar{h}(x)$, we can compare the deviation of $h(x)$ from $\bar{h}(x)$ using a loss function $\ell(h, x, \bar{h}(x))$. The choice of the loss function depends on the specific problem and data, but common options include the cross-entropy loss for classification and the mean squared error loss for regression. In this work, we focus on neural classification models and leave other models (*e.g.*, generative models) to future work. Therefore, we write $G_x = \arg \max \bar{h}(x)$ to denote the labelled category that $x$ belongs to, and $h(x)$ denotes the logits output by model $h$.

**Adversarial Examples and Robustness in Classification**   Adversarial examples are inputs that are carefully crafted by adding human imperceptible perturbation to normal inputs to trigger wrong predictions (Szegedy et al., 2014; Kurakin et al., 2017a). These perturbations can be hardly perceptible to the human eye (Carlini et al., 2019). The existence of an adversarial example can be defined as the presence of two inputs that are nearly identical but are assigned different classifications by the model. Formally, an adversarial example exists if and only if the following is satisfied.

$$\exists\, x_1, x_2 \in \mathcal{X}.\ \ d(x_1, x_2) \leq \epsilon\ \wedge\ \ \arg\max h(x_1) \neq \arg\max h(x_2) \tag{1}$$

where $d(x_1, x_2)$ denotes a distance measure between the two inputs, and this distance needs to be smaller than a threshold $\epsilon$. Note that the distance function can be defined in a variety of ways (*e.g.*, based on some $L^p$-norm or the degree of rotation). The robustness of a neural network model qualifies its ability to maintain its prediction in the presence of small perturbations, which is expressed in Equation (2).

$$P_{x_1 \sim \mathcal{D}} \bigg( P\Big( \arg\max h(x_1) \neq \arg\max h(x_2) \mid d(x_1, x_2) \leq \epsilon \Big) \leq \kappa \bigg) \tag{2}$$

where $\kappa$ is a constant threshold in the range $[0, 1]$ and $\mathcal{D}$ is the distribution of input data. When $\kappa = 0$, it is known as deterministic robustness (Madry et al., 2018; Pang et al., 2022; Li et al., 2023). Otherwise, it is commonly known as probabilistic robustness (Zhang et al., 2023; Li et al., 2023).

State-of-the-art robustness-aware training methods can be broadly categorized into adversarial training (Ganin et al., 2016), certified training (Singh et al., 2019), and others. Specifically, the goal of adversarial training and certified training is captured below by Equation (3) and (4) respectively.

$$\min_h\ \ \mathrm{E}_{x \sim \mathcal{D}} \left[ \max_{d(x,t) \leq \epsilon} \ell\big(h, t, G_t\big) \right] \tag{3}$$

$$\min_h\ \ \mathrm{E}_{x \sim \mathcal{D}} \left[ \sup_{d(x,t) \leq \epsilon,\ c \neq G_t} \Big( \ell(h, t, G_t) - \ell(h, t, c) \Big) \right] \tag{4}$$

where $\ell$ is a loss function, $G_t$ is the ground truth prediction for $t$. Intuitively, adversarial training approximates the worst loss that can be induced by a perturbation and tunes model $h$ to minimize this loss, whereas certified training looks for an upper bound of the loss and tunes the model to minimize it. Known limitations of adversarial training include a trade-off between accuracy and robustness and the lack of guarantee of robustness against evolving adversarial attacks. Certified training, on the contrary, guarantees robustness, but often leads to a significantly dropped accuracy. These limitations call for a technique that certifies robustness as well as maintains accuracy. More details on how adversarial training and certified training work are presented in the Appendix A.

**Problem Definition**   Achieving certified probabilistic robustness whilst maintaining high accuracy is the goal of this work. Unlike deterministic robustness, probabilistic robustness allows a small number of exceptions within the vicinity of a sample to have different labels, which makes it much more achievable in practice. Furthermore, certified probabilistic robustness provides theoretical guarantees for the model performance when faced with adversarial inputs, which could be useful for system-level decision-making. In practice, it is often sufficient to keep the probability of undesirable events from occurring sufficiently low (ISO, 2014).

However, achieving both (high) certified probabilistic robustness and accuracy is challenging. This research aims to provide a solution that ensures accuracy on clean test data, robustness against leading adversarial attacks like AutoAttack (Croce & Hein, 2020b), and certified probabilistic robustness for test samples. Moreover, the solution expects efficiency during both training and inference, especially for larger models, and seamless integration with existing architectures and frameworks.

## 3   OUR METHOD

Our method consists of two complementary parts. The first is a training method that aims to improve probabilistic robustness by minimizing the variance across the vicinity, illustrated in Figure 1. The second is an inference method that aims to establish certified robust prediction for a given sample.

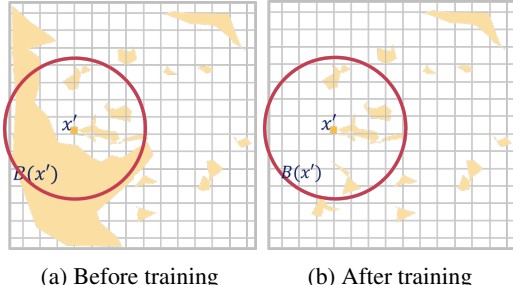

(a) Before training    (b) After training

Figure 1: Intuition on how our training method works. Suppose $x'$ is an adversarial example that is incorrectly predicted. The yellow shadow represents other adversarial examples. We achieve probabilistic robustness in vicinity $B$, say $P(t \text{ correctly predicted } \mid t \in B(x')) \geq 1/2$, by minimizing the variance within the vicinity which allows occasional misprediction.

### 3.1 Variance-Minimizing Training

To obtain an accurate and likely probablistically robust model, we minimize the variance among model outputs for inputs within the same vicinity, together with empirical risk minimization (ERM (Vapnik, 1999)). This training can be formulated as a Pareto optimization problem whose objective is as follows.

$$\min_h \quad \left\{ \mathrm{E}_{x \sim \mathcal{D}} \left[ \mathrm{E}_{d(x,t) \leq \epsilon} \; \ell(h, t, G_t) \right], \quad \mathrm{E}_{x \sim \mathcal{D}} \left[ \mathrm{Var}_{d(x,t) \leq \epsilon} \; \ell(h, t, G_t) \right] \right\} \tag{5}$$

where the first term is essentially the objective of ERM (Vapnik, 1999), and the second term, variance of individual losses, is the novel part.

Implementation-wise, during each training step where we are given a training sample $x$, we first sample a fixed number of (perturbed) inputs within the vicinity of $x$. Then, we use the neural network to make a prediction on each of the samples. Next, we compute the individual loss for each sample against the label of $x$ independently. We then calculate the mean and standard deviation of these individual losses. Finally, we use a weighted sum of the mean and standard deviation as the final loss (*i.e.*, $\mathrm{E}_{d(x,t) \leq \epsilon} \; \ell(h, t, G_t) + \lambda \mathrm{Var}^{\frac{1}{2}}_{d(x,t) \leq \epsilon} \; \ell(h, t, G_t)$) to back-propagate gradients and update the parameters of the neural network with the given learning rate. This iterative optimization can be governed by stochastic gradient descent (SGD) or other optimizers.

Note that the loss function combines mean minimization and variance minimization, with a weighting factor $\lambda$ determining the importance of each component. Furthermore, we use the square root of the variance term, allowing a linear combination of mean and standard deviation for the loss back-propagation. Intuitively, minimizing the variance allows us to improve the model's robustness without depending on any specific adversarial attacking methods. Instead, we improve model robustness by minimizing the spread (standard deviation) of model prediction alongside the traditional ERM method. Random sampling is adopted and the adversarial attack in each training step is avoided. In the ideal case, if the sample $x$ is correctly predicted by a model and the predictions of *any pair* of samples in the vicinity are the same, this model achieves deterministic robustness in that vicinity. Likely in practice, by minimizing the variance of the loss within the vicinity of each $x$, many samples within the vicinity have the same (correct) prediction. In this way, the proposed training differs from existing adversarial training methods which either rely on pre-computed adversarial examples (Miyato et al., 2019) or adversarial examples generated during training (Zhang et al., 2019a) (often paying a high training cost). A more formal argument on why minimizing variance leads to improved robustness is present in the Appendix B.

Note that, both terms in Objective (5) are crucial. Variance in the data represents the difference between individual observations. High variance means that the observations are scattered, while low variance means they are tightly clustered around the mean. Namely, by decreasing prediction variance, the model becomes more robust whereas focusing only on reducing the mean, as seen with data augmentation (Wen et al., 2021), may leave outliers unpredictable, potentially leading to adversarial examples. On the other hand, minimising the variance cannot be the sole objective, as doing so would result in the model consistently making incorrect predictions for all samples.

Consequently, the hyperparameter $\lambda$ must be reasonable, *i.e.*, an overly-large $\lambda$ leads to reduced model accuracy; and a small $\lambda$ makes the model less robust.

## 3.2 INFERENCE AND CERTIFICATION

Unlike certified training, our training method itself does not provide robustness certification. The second part of our approach is an inference algorithm which aims to provide certified probabilistic robustness when possible. According to Equation (2), to establish certified probabilistic robustness, we must show that there is a guaranteed upper bound on the probability of adversarial examples, *i.e.*, some threshold $\kappa$. Intuitively, we would like to know for sure that among all the samples within the vicinity around an input, at least $1 - \kappa$ of them are not adversarial examples.

Our inference method is designed to serve two purposes, *i.e.*, providing a prediction, and offering certified probabilistic robustness. The general idea is for any $x \in \mathcal{X}$ and model $h$,

$$h^{\dagger}(x) = \mathrm{E}_{d(x,t) \leq \epsilon} \, h(t) \tag{6}$$

where the superscript $\dagger$ is used to distinguish our prediction method from the vanilla inference. What matters more is how this inference provides a robustness certificate during inference. Specifically, when deciding a prediction for $x$, we sample sufficiently many samples in its vicinity and make the prediction based on the majority of the predictions. Thus, a model would only make mistakes when more than half of the sampled samples are predicted wrongly. To determine whether the proportion of minority prediction is bounded by $\kappa$, a statistical hypothesis test is conducted, which tells us either to accept or reject the hypothesis that the probability of minority prediction in this vicinity is lower than $\kappa$. We only generate a robustness certificate if the corresponding hypothesis is accepted.

Particularly, the exact binomial test (Blitzstein & Hwang, 2019) is adopted as the hypothesis test. The binomial test assesses a hypotheses about population proportions for binary variables based on sampled observations. It evaluates whether the proportion of a value in a binary variable is less than, greater than, or not equal to a specific value. To evaluate the hypothesis on whether a certain class of prediction around an input accounts for more than $\kappa$ (*e.g.*, 10%), we first express the probability of observing the given class occurrence as

$$P(Z \geq z \mid p = \kappa) = 1 - \sum_{i=0}^{z-1} \binom{n}{i} (\kappa)^i (1 - \kappa)^{n-i} \tag{7}$$

where the number of observed occurrences of this class in the sample is a random variable $Z$ evaluated at $z$. $n$ is the sample size; $p$ is the claimed population proportion (in this case, $\kappa$); $\binom{n}{i}$ is the binomial coefficient, which calculates the number of ways to choose $i$ items from a set of $n$ items.

Then, if the resulting probability is less than a pre-determined significance level (*e.g.*, $\alpha = 0.05$), we reject the null hypothesis that the proportion of occurrences is greater than or equal to 10% and conclude that it is lower. Otherwise, we fail to reject the null hypothesis and conclude that there is not enough evidence to suggest that the proportion is lower than 10%. Note that, failing to reject the null hypothesis does not indicate accepting it. Thus, we next perform both left-tail, *i.e.*, $P(Z \leq z)$, and right-tail binomial tests to ensure that we can either reject that the probability is less than $\kappa$ or greater than $\kappa$. Still, we might be able to reject neither because the sample size $n$ is not large enough. To this end, we use sequential sampling (Wald, 1945) such that the sampling continues until we manage to reject either hypothesis. This provides certainty as to whether the prediction on the test case is certified as robust or not. Details of the algorithm are presented in the Appendix B.

## 4 EXPERIMENT

In this section, we report the experimental results on applying our method. The experiment focuses on the following research questions: (RQ1) is our method effective in achieving robustness whilst maintaining accuracy; (RQ2) how effective is our method in defending adversarial attacks; (RQ3) how efficient is our approach; and (RQ4) how do the hyper-parameters impact the performance of our approach. To answer the first three RQs, we compare our method against eight existing methods based on common benchmark datasets. The eight baselines are empirical risk minimization (ERM) (Vapnik, 1999), data augmented training (DA) (Shorten & Khoshgoftaar, 2019), PGD-based

Table 1: Performance comparison on $L^\infty$-norm based perturbation.

| Approach | Standard Accuracy | | | | Certified Robustness Rate | | | | Certified Robust Accuracy | | | |
|---|---|---|---|---|---|---|---|---|---|---|---|---|
| | CIFAR-100 | CIFAR-10 | SVHN | MNIST | CIFAR-100 | CIFAR-10 | SVHN | MNIST | CIFAR-100 | CIFAR-10 | SVHN | MNIST |
| ERM | **81.03** | **94.85** | 94.44 | 99.37 | 9.28 | 1.25 | 52.72 | 26.01 | 4.52 | 1.25 | 51.04 | 24.96 |
| DA | 78.27 | 94.21 | 94.69 | **99.42** | 15.04 | 81.08 | 82.08 | 85.23 | 6.15 | 76.07 | 82.01 | 84.12 |
| PGDT | 64.35 | 84.38 | 91.19 | 99.16 | 57.07 | 87.07 | 87.89 | 94.65 | 32.93 | 82.90 | 86.68 | 94.63 |
| TRADES | 62.55 | 80.42 | 86.16 | 99.10 | 59.27 | 88.54 | 87.89 | 94.76 | 38.85 | 78.80 | 84.76 | 94.61 |
| MART | 63.68 | 81.54 | 90.20 | 98.94 | 58.79 | 78.90 | 85.23 | 94.13 | 49.37 | 72.21 | 78.82 | 94.09 |
| RS | 56.87 | 89.45 | 88.35 | 97.16 | 60.38 | 90.00 | 76.29 | 87.15 | 47.50 | 87.98 | 70.64 | 86.29 |
| IBP | 39.45 | 48.40 | 73.09 | 97.78 | 49.34 | 54.70 | 61.94 | 89.18 | 29.20 | 40.00 | 57.26 | 88.51 |
| PRL | 64.89 | 93.82 | 92.00 | 99.32 | 56.71 | 90.71 | 93.11 | 96.03 | 50.77 | 90.63 | 91.07 | 95.01 |
| Ours | 65.56 | 94.23 | **94.79** | 99.32 | **62.05** | **95.08** | **93.15** | **97.80** | **52.07** | **91.75** | **92.81** | **96.80** |

$\kappa = 10^{-2}, 1 - \alpha = 0.99$; $L^\infty$ bound at 0.3 for MNIST, and 8/255 for CIFAR-10, CIFAR-100, or SVHN.

adversarial training (PGDT) (Madry et al., 2018), TRADES (Zhang et al., 2019a), MART (Wang et al., 2020), Randomized Smoothing (RS) (Cohen et al., 2019), IBP (Shi et al., 2021) (which is a certified training method), and PRL (Robey et al., 2022) (which is a training method for probabilistic robustness, with no guarantee). We adopt existing open-source implementations of each method.

Four popular classification datasets are adopted, *i.e.*, MNIST (LeCun et al.), SVHN (Netzer et al., 2011), CIFAR-10 (Krizhevsky et al., 2009), and CIFAR-100 (Krizhevsky et al., 2009). The original training set of each dataset comprises a minimum of 50,000 samples, which are partitioned into training and validation sets with a ratio of 8:2. We adopt multiple standard model architectures to train the classifiers on the above-mentioned datasets. Further details on the training parameters, the perturbations and model architectures are present in the Appendix C.1. Our implementation, trained models, and supplementary materials are available at `https://github.com/soumission-anonyme`.

**RQ1: Is our method effective in achieving robustness whilst maintaining accuracy?** To answer this question, we evaluate our method and baseline methods using three metrics, *i.e.*, standard accuracy, certified robustness rate, and certified robust accuracy. Intuitively, standard accuracy measures the probability that the model's prediction is correct for an input from the natural distribution; certified robustness rate measures the probability that a prediction is certified to be robust; certified robust accuracy measures the probability that a prediction has certified robustness and is correct. Note that for the baseline methods that do not inherently report probabilistic certified robustness, we run the exact same binomial test as in our method to verify their certified robustness rate. The results based on $L^\infty$-norm based perturbation are shown in Table 1.

In terms of standard accuracy, our method exhibits a reasonably small sacrifice on standard accuracy (whilst achieving robustness), compared to most of the existing methods. On the CIFAR-10, SVHN, and MNIST datasets, our method has a slight decreased accuracy in comparison to ERM, with a maximum reduction of less than 0.7% and an average of 0.1%, which is close to that of DA. On CIFAR-100, although there is a noticeable decrease in accuracy compared to ERM, our method still ranks the second-best method, only surpassed by DA. Noticeably, adversarial training results in a minimum 8.35% drop in accuracy, certified training usually leads to over 40% accuracy drop, and randomised smoothing results in a 10.31% accuracy drop. These baselines sacrifice standard accuracy significantly for robustness expectedly. The reason is apparent when we consider their details, *e.g.*, randomised smoothing introduces Gaussian noise during the training process to improve the model's robustness to perturbations, which can inadvertently push some of the original samples farther away from their true labels, leading to a reduction in accuracy.

In terms of certified robust accuracy, it can be observed that our method has the highest certified robust accuracy on all four datasets, with an average of 83.36% (*i.e.*, we are able to certify that the prediction is probabilistically robust for most of the time). In comparison, PRL is the best-performing baseline method, with an average certified robust accuracy of 81.87%. Furthermore, comparing the results on the different datasets, we observe that our method outperforms PRL more when the dataset is more complex. Additionally, the average certified robust accuracy of the best adversarial training method (*i.e.*, PGDT) is 74.29%, which is 11.42% lower than ours. Randomized smoothing and IBP yield even worse results (*e.g.*, at 73.10% and 53.74% respectively), although they still outperform vanilla training (*i.e.*, ERM) whose certified robust accuracy is only 20.44%.

Table 2: Performance comparison on other kinds of perturbations.

| Approach | Translation ($\pm 0.3$) | Rotation ($\pm 35°$) | Affine | Scale ($\pm 0.3$) |
|---|---|---|---|---|
| ERM | 92.85 | 93.95 | 92.85 | 93.01 |
| DA | 93.97 | 92.76 | 92.75 | 93.23 |
| PGDT | 64.35 | 74.23 | 61.34 | 69.46 |
| TRADES | 68.72 | 74.89 | 64.56 | 74.45 |
| MART | 73.48 | 81.35 | 74.23 | 68.49 |
| RS | 87.25 | 86.29 | 82.51 | 87.58 |
| IBP | 46.52 | 46.94 | 44.41 | 47.80 |
| PRL | 90.68 | 91.74 | 89.23 | 90.92 |
| Ours | **93.28** | **94.15** | **93.28** | **93.25** |
| $\kappa = 10^{-2}, 1 - \alpha = 0.99$ | | | | |

Table 3: Comparing the defence success rates against AutoAttack (Croce & Hein, 2020b).

| Approach | Defence Success Rate | | | |
|---|---|---|---|---|
| | CIFAR-100 | CIFAR-10 | SVHN | MNIST |
| ERM | 0.01 | 0.00 | 2.72 | 0.01 |
| DA | 0.03 | 0.00 | 2.08 | 5.23 |
| PGDT | 31.48 | 40.90 | 44.89 | 94.65 |
| TRADES | 33.05 | 44.35 | 54.89 | 94.76 |
| MART | 32.43 | 38.10 | 45.23 | 94.13 |
| RS | 9.25 | 0.00 | 56.29 | 87.15 |
| IBP | 29.33 | 37.10 | 61.94 | 89.18 |
| PRL | 0.00 | 0.71 | 3.11 | 26.03 |
| Ours | **53.05** | **88.08** | **92.15** | **97.8** |
| $L^{\infty}$ bound at 0.3 for MNIST, and 8/255 for CIFAR or SVHN. | | | | |

In terms of certified robustness rate, our method achieves the best performance on certified robustness rate, with an average value of 87.02%, which is 3.42% and 36.42% higher than PRL and certified training, respectively. The baselines that achieve higher standard accuracy than ours, *i.e.*, ERM and DA, have significantly lower certified robustness rate, with an average rate of 22.32% and 65.86%, respectively.

**More than $L^p$ transformation.** While the existing robustness certification methods such as certified training and randomized smoothing primarily focus on $L^p$ transformations of images (Shi et al., 2021), as presented in Table 1, we are also interested in the certified robustness on other transformations, such as translation, rotation, affine, and scaling. In our experiments, we randomly perturb the input within the given range of each transformation and report the corresponding certified robust accuracy. Similar to the previous experiments, we apply the exact binomial test to verify the robustness of the model obtained by different training algorithms with respect to non-$L^p$ norm transformation. The hyper-parameters $\kappa$ and $\alpha$ are set as $10^{-2}$ and $0.01$, respectively. We present the results on CIFAR-10 in Table 2 and similar results are obtained on other datasets. It can be observed that our method consistently achieves the highest certified robust accuracy across all transformations, surpassing the threshold of 93.49%. The second highest is DA, with all results above 92.23%. This is likely because, rotation, translation, and scaling are frequently used in data augmentation. Remarkably, ERM achieves the third-highest certified robust accuracy, which can be attributed to the inherent robustness of convolutional layers to these non-$L^p$ transformations. This robustness is due to their ability to capture and extract local patterns and spatial relationships in images through shared weights, local receptive fields, and spatial pooling operations (He et al., 2016). In addition, PRL has slightly worse performance than our method, *i.e.*, by 3.05%.

> ***Answer to RQ1***: Our approach achieves the highest certified robust accuracy, highest certified robustness rate, and has a <1% accuracy drop compared to standard training).

**RQ2: How effective is our method in defending adversarial attacks?** Two methods may achieve the same probabilistic robustness but have different behaviours when facing adversarial attacks, *i.e.*, one may be easier to attack as those few adversarial samples may be easier to identify by existing adversarial attacking methods. We thus adopt the state-of-the-art method, AutoAt-

tack (Croce & Hein, 2020b), to evaluate the effectiveness of our method and baselines in terms of how effective the respective models can defend against adversarial attacks. AutoAttack is an ensemble of different PGD attacks that is parameter-free, computationally affordable, and user-independent, making it an effective tool for assessing adversarial accuracy. We systematically apply AutoAttack to each of the models trained with different methods with the same attacking configuration (*i.e.*, same budget for perturbation and time). The results are as shown in Table 3.

The results suggest that our proposed method consistently achieves a higher defence success rate than any other approach across all four datasets. Specifically, our method achieves an impressive defence success rate of 82.77% on average, surpassing the best performance of adversarial training, *i.e.*, TRADES, whose defence rate is 56.76%. Apart from adversarial training, IBP achieves the highest defence success rate at an average of 54.39%. It is worth noting that although PRL exhibits relatively high certified robust accuracy, second only to our method, their resilience against adversarial attacks is significantly low, with a defence success rate close to 0 on CIFAR-10 and CIFAR-100. Intuitively, the reasons why our method is more successful in defending AutoAttack are twofold. First, existing adversarial training methods are based on specific attacks and are thus fragile to new attacks, whereas our variance-minimizing training is not. Second, the inference part of our method forces an attacker to generate "robust" adversarial samples (Athalye et al., 2018b) (*i.e.*, a region where more than half of the samples are adversarial) to attack successfully, which is much more challenging given the scarcity of adversarial examples after our variance-minimizing training.

In addition, we evaluate our method and baselines with a comprehensive list of 25 adversarial attacks from the TorchAttacks library (Kim, 2020) on the CIFAR-10 model and compare the defence success rates. The results are consistent with the above (*i.e.*, our method always has a higher defence success rate than baselines). The detailed results are shown in Table 9 of the Appendix C.2.

Overall, our method demonstrates effectiveness in defending against a wide range of adversarial attacks. We remark while it is true that probabilistic robustness weakens deterministic robustness, this weakening is perhaps acceptable as evidenced by the above-mentioned results. That is, our approach does make adversarial attacking much harder.

> ***Answer to RQ2***: Our approach has the highest defence success rate against AutoAttack.

**RQ3: How efficient is our approach?**  To answer this question, we measure the training and inference time of our method and the baseline approaches. The results on MNIST are shown in Table 4 and similar results are observed on other models. For inference, the time is collected on the whole testing set comprised of 10,000 samples.

Table 4: Comparison of overhead for our training on MNIST dataset with 300 epoch training time. The training cost of all approaches is collected using a single NVIDIA RTX 2080 Ti GPU.

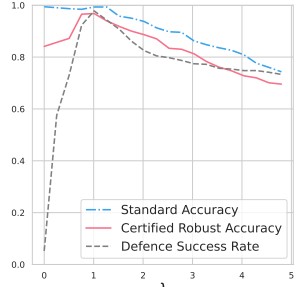

| Approach | Training time (sec) | Inference time (sec) |
|---|---|---|
| ERM | $4.9 \times 10^2$ | 27 |
| DA | $8.9 \times 10^3$ | 27 |
| PGD | $9.9 \times 10^3$ | 27 |
| TRADES | $9.9 \times 10^3$ | 27 |
| MART | $9.9 \times 10^3$ | 27 |
| RS | $5.8 \times 10^4$ | $2.7 \times 10^5$ |
| IBP | $2.1 \times 10^5$ | 27 |
| PRL | $2.8 \times 10^4$ | $2.7 \times 10^3$ |
| Ours | $9.6 \times 10^3$ | $4.7 \times 10^3$ |

Figure 2: Adjusting hyperparameter $\lambda$ makes the model converge with different performance. The experiment is run on MNIST with an $L_\infty$ bound of 0.3.

For training efficiency, we can observe that compared to methods designed to certify robustness, *i.e.*, IBP and PRL, our method demonstrates significantly higher training efficiency, being 21.93 and 2.89 times faster, respectively. Our method has a similar training cost to data augmentation and adversarial training, with a total training time around 10 thousand seconds. These findings indicate that our approach is highly efficient and practical for training deep neural networks with robustness guarantees. On inference efficiency, as ERM, DA, and the adversarial training methods do not provide robustness certification, their inference is more efficient (similar to that of IBP as shown

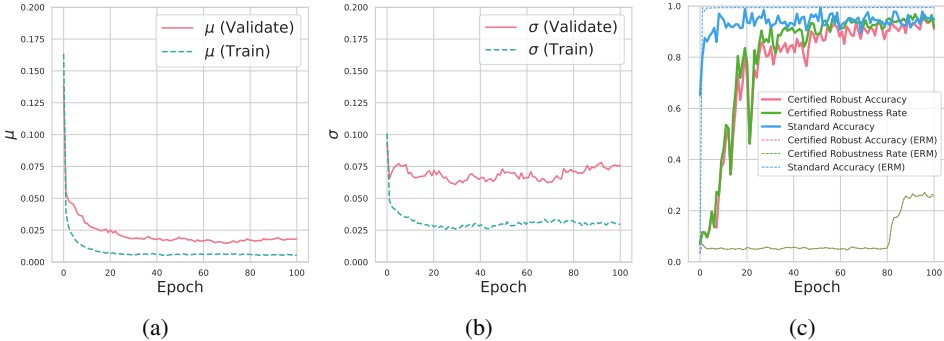

Figure 3: (a) & (b) Illustrations of loss convergence of the mean term and variance term in Objective (5). (c) Illustration of model performance in terms of certified robust accuracy, certified robustness rate as well as standard accuracy. The illustrated figures are from experiments on MNIST.

below). We thus focus on comparing our method with certified training algorithms, *i.e.*, RS, IBP, and PRL. From Table 4, it can be observed that IBP takes the least inference time as it only requires a single forward propagation on the input to obtain the predictions and certification results. In contrast, the other three methods provide certification by predicting a large number of samples around the input. Our inference can be considered reasonably efficient, as it has the same order of magnitude as PRL and is two orders of magnitude faster than RS. This is mainly attributed to our method using sequential sampling to reduce processing time. That is, sequential sampling allows for decisions to be made based on observed data at each step (Wald, 1945) and is known to reduce required sample sizes while maintaining statistical correctness due to its adaptability (Mead, 1990). In comparison to fixed-size sampling, sequential sampling may lead to increased efficiency (Chernoff, 1959).

We further show the process of performance coverage in Figure 3 on MNIST. It can be observed that our method converges efficiently on correctly predicting unperturbed samples, and convergence on the perturbed samples is slightly delayed, as illustrated in Figure 3(c).

> ***Answer to RQ3***: Our approach has a training cost similar to data augmentation and adversarial training, and is much more efficient than certified training.

**RQ4: How do the hyper-parameters impact the performance of our approach?** We conduct an ablation study to assess the effect of the importance factor $\lambda$ in our method, measured using standard accuracy, certified robust accuracy, and defence success rate against AutoAttack. The value of $\lambda$ ranges from 0 to 5, with an interval of 0.25. Figure 2 presents the trend of changes in model performance for MNIST, which is representative of other results. Note that a value of $\lambda$ close to 1 yields the best performance. When the value of $\lambda$ decreases, the contribution of the proposed variance-minimization term decreases. If $\lambda$ is too small, *i.e.*, close to 0, the training process becomes similar to data-augmented training with random perturbation, resulting in a drop in the certified robust rate and thus decreasing the certified robustness accuracy. On the other hand, if the loss function excessively emphasizes the variance term with a large value of $\lambda$, it can lead to a decrease in standard accuracy and further impact the certified robust accuracy. Additionally, the defence success rate also decreases by about a quarter when varying $\lambda$ from 1 to 5.

## 5   CONCLUSION

We present an approach that improves the robustness of neural networks against adversarial examples. Our approach includes a training method that minimizes both the mean and variance of the loss in prediction and an inference method that provides probabilistic-certified robustness. Through theoretical analysis, we have shown that minimizing variance is the upper bound of the probability of adversarial examples and that higher quantile accuracy leads to over 91% certified robust accuracy. Our experimental results on standard benchmark datasets show that our method achieves higher defence success rate and certification rate compared to the state-of-the-art while sacrificing less standard accuracy.

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

# A APPENDIX: ADVERSARIAL TRAINING AND CERTIFIED TRAINING

In the past decade, various methods have been proposed to improve the robustness of models, which have been reviewed in recent studies (Silva & Najafirad, 2020; Li et al., 2023). State-of-the-art training methods can be broadly categorized into adversarial training (Ganin et al., 2016), certified training (Singh et al., 2019), and a number of other approaches. In the following, we extend our discussion in Section 2.

**Adversarial training** The general idea of adversarial training is given in Equation (3). It aims at improving a model's empirical robustness, and there can be many variants of the method. For instance, the loss function $\ell$ can can vary in type, (*e.g.*, the 0-1, cross-entropy, or squared loss). A critical part of adversarial training is how they search for adversarial inputs within the vicinity of the training samples. Goodfellow et al. introduce a fast gradient sign method (FGSM) to generate adversarial inputs (Goodfellow et al., 2015). Adversarial training with FGSM significantly improves a model's robustness against adversarial samples generated through FGSM. Various other adversarial attacking methods are adopted for adversarial training as well. Among them, Projected Gradient Descent (PGD (Madry et al., 2018)) based adversarial training is shown to be the most effective, in various domains, including image classification and reinforcement learning. In the context of large-scale image classification tasks, an ensemble adversarial training method further improves robustness through utilizing adversarial examples generated from multiple pre-trained models (Kurakin et al., 2017a).

Despite the advancements made in adversarial training over the years, improving model robustness remains an open problem. This is partly due to the challenge posed by the trade-off between standard accuracy and robustness (Tsipras et al., 2019). To this end, TRADES is proposed to balance this trade-off with a regularization term based on Kullback-Leibler (KL) divergence between the model's output on clean inputs and adversarial inputs (Zhang et al., 2019a). This approach has achieved state-of-the-art performance on several benchmark datasets, including CIFAR-10. Nevertheless, a 15% accuracy drop is still observed.

More importantly, a significant limitation of adversarial training is that it does not certify a model's robustness against adversarial attacks (Balunovic & Vechev, 2020). This lack of certification implies that the robustness of a model cannot be guaranteed, particularly as new and sophisticated adversarial attacking methods are being developed (Athalye et al., 2018a). For instance, it has been shown that a model trained through adversarial training remains vulnerable to new threats such as adaptive adversarial attacks (Liu et al., 2019a; Tramer et al., 2020). This limitation highlights the need for techniques that can provide certified robustness, *i.e.*, a guarantee that the model is robust no matter what adversarial attacks are conducted.

**Certified Training** Certified training aims to train models that are certified to be robust (Vaishnavi et al., 2022). The idea is to soundly approximate the effect of any adversarial attack method and optimize the parameters of model $h$ so that the effect of any adversarial attack method is kept within a certain bound such that deterministic robustness is guaranteed (Li et al., 2023).

To soundly approximate the effect of any adversarial attack method, existing certified training methods use neural network verification techniques to soundly approximate the worst loss that can be induced by any perturbation within the vicinity of each training sample. If the label remains the same in the presence of such worst loss, the model is certified to be robust with respect to the sample. Note that after years of development, many neural network verification techniques have been proposed, *e.g.*, (Zhang et al., 2018; Singh et al., 2019; Balunovic & Vechev, 2020).

Certified training methods however suffer from multiple shortcomings. First, they are computationally expensive. Although there has been a lot of development in neural network verification techniques, it is perhaps fair to say that such methods are still limited to relatively small neural networks. Given that certified training requires verifying the neural network robustness against each and every training sample, certified training is limited to small neural networks as of now. Second, existing certified training methods often result in a significant drop in the model's clean accuracy, *i.e.*, accuracy on clean, non-adversarial inputs (Cohen et al., 2019; Raghunathan et al., 2018). The best clean accuracy achieved by certified training is typically 70% of that from adversarial training on the CIFAR-10 dataset (Tsipras et al., 2019; Shi et al., 2021). Such dramatic accuracy drop makes

their application in real-world systems rare as of now. Lastly, existing certified training methods usually only work for robustness defined based on the $L^p$ norms or in rare cases, simple transformation such as image rotation (Cohen et al., 2019).

## B  APPENDIX: DETAILS ON THE PROPOSED METHOD

In the following, we present details of the intuition and implementation of our method, including both training and inference. For training, we want to show why models with lower variance among nearby predictions are more robust.

**Proposition B.1.** *If two distributions with the same mean have different variances, where the variance of one is less than the other, then for any quantile level $q$ in the range $0 < q < 1$, the upper bound of the $q$-th quantile of the distribution with the lower variance is less than the upper bound of the $q$-th quantile of the distribution with the higher variance.*

*Proof.* We start with Chebyshev's inequality. Chebyshev's inequality provides an upper bound on the tail probabilities of a random variable based on its variance. Let $Z$ (integrable) be a random variable with finite expected value $\mu$ and finite non-zero variance $\sigma^2$. Then for any real number $\lambda > 0$,

$$P\big(|Z - \mu| \geq \lambda\sigma\big) \leq \frac{1}{\lambda^2} \tag{8}$$

which states that for any probability distribution, the proportion of data within $\lambda$ standard deviations of the mean is at least $1 - 1/\lambda^2$, and we can further derive:

$$P\big(|Z - \mu| \geq \lambda\sigma\big) = P\big(Z - \mu > \lambda\sigma\big) \leq \frac{1}{\lambda^2}$$
$$P\big(Z \leq \lambda\sigma + \mu\big) \geq 1 - \frac{1}{\lambda^2} \tag{9}$$

Let $\lambda\sigma = \mu - z$, we can have the inequality flipped like:

$$P\big(Z \leq z\big) \geq 1 - \frac{\sigma^2}{(z - \mu)^2} \tag{10}$$

For any given $z$, When the variance $\sigma^2$ decreases, the lower bound for $P\big(Z \leq z\big)$ increases. Hence, minimizing the variance is essentially reducing the probability of examples far away from the mean. □

By minimizing the variance across the perturbation space for each sample in the training set, probabilistic robustness can be improved. Then, we show the detailed and practical training algorithm in Algorithm 1.

Specifically, in line 7, the sampling follows a uniform distribution because the vicinity is usually defined to be a small region and there is hardly a reason why some samples have higher probability than others. Here, we use $B$ to represent the vicinity, similar to the case in Figure 1. Formally, it is equivalent to the distance notation used throughout in this work, *e.g.*, in Equation (3), and for all $x \in \mathcal{X}, B(x) = \{t \mid d(x,t) \leq \epsilon\}$. Generally, vicinity of $x$ can be defined to be some $L^p$ norm of $x$ (where $p = 0, 1, 2, \infty$) (Kurakin et al., 2017a), or domain-specific label-preserving transformations (*e.g.*, tilting and zoom in/out) (Athalye et al., 2018b; Bhattacharya & Gupta, 2019). Specifically, when a vicinity is characterised by a distance function $d$ and a predefined threshold $\epsilon$, common notations of distances include

$$d(x_1, x_2) = \|x_1 - x_2\|_p, \quad \text{(Additive in } L^p \text{ norm), or}$$
$$d(x_1, x_2) = \begin{cases} |\epsilon'|, & \text{if } f_{\text{transform}}(x_1, \epsilon') = x_2, \\ \epsilon + 1, & \text{otherwise} \end{cases} \tag{11}$$

where the transform function mapping from $\mathcal{X}$ to $\mathcal{X}$ can be understood as a specific transformation (*e.g.*, whether an image is rotated or horizontally shifted) and its parameters ($\epsilon'$, *e.g.*, the *degree* of rotation).

---

**Algorithm 1** Variance-Minimizing Training

---

1: **Input:** Training data $\{\,(x_i, G_{x_i}) \mid i = 1, 2, \ldots, k; x_i \in \mathcal{X}\,\}$, predefined vicinity $B$, network architecture $h_\theta$ parametrized by $\theta$, step size $\eta$, sample size $n$, batch size $m$, and $\lambda$.
2: **Initialization:** Standard random initialization of $h_\theta$
3: **Output:** Robust network $h_\theta$
4: **repeat**
5:    Uniformly sample $\{\,(x_i, G_{x_i}) \mid i = 1, 2, \ldots, m\,\}$, a minibatch of training data where $m < k$
6:    **for** $i = 1, 2, \ldots, m$ **do**
7:       Draw $\{\,t_j \mid j = 1, 2, \ldots, n\,\} \sim \mathcal{U}(B(x_i))$ where $\mathcal{U}$ is a uniform distribution
8:       **for all** $j = 1, 2, \ldots, n$ **do**
9:          $u_j \leftarrow l_{\text{Cross-entropy}}(h_\theta(t_j), G_{x_i})$
10:       **end for**
11:       $\mu_i \leftarrow \sum_{j=1}^{n} u_j / n$
12:       $\sigma_i \leftarrow (\sum_{a=1}^{n} \sum_{b=1}^{n} (u_a - u_b)^2 / n)^{1/2}$
13:    **end for**
14:    $\theta \leftarrow \theta - \eta \sum_{i=1}^{m} \nabla_\theta [\mu_i + \lambda\, \sigma_i] / m$
15: **until** convergence

---

In line 14, the loss function (mentioned in Section 3.1) is constructed by a weighted sum of mean and standard deviation with importance factor $\lambda$.

For inference, a feasible step-by-step implementation of this idea is presented in Algorithm 2. Specifically, in lines 7-21, the binomial test is described. The level of statistical significance is determined by $\alpha$. As $\alpha$ decreases, the statistical significance increases, which means that the certification is less likely to result in a false positive. Additionally, those cases that are not certified as robust have a lower likelihood of being false negatives. Although $\kappa$ and $\alpha$ are typically selected within the range of $10^{-1}$ to $10^{-4}$, decreasing both values, *i.e.*, $\kappa \to 0$ and $\alpha \to 0$, can make the certification more reliable. In line 19, we can see that the stop criterion for collecting data is the probability of either the right or left tail crossing a predefined false positive rate. We make a decision based on which tail has crossed the threshold and certify the prediction as either robust or non-robust accordingly.

**Theorem B.2.** *Let $x$ be a sample. If Algorithm 2 returns that $x$ has certified robustness,* i.e., $p_{right} < \alpha$, *then the probabilistic robustness of $x$ is greater than $1 - \kappa$ is satisfied.*

## C   APPENDIX: DETAILS ON EXPERIMENTS

### C.1   EXPERIMENT SETUP

The details of the model architectures are summarized in Table 5. These architectures all have been studied by existing robustness improving methods, as shown in the *Works* column. In short, the model size ranges from 378,562 parameters for the small CNN7 model, to 11,689,512 parameters for the more complex ResNet-18 model.

Table 5: Details of model architectures

| Model | # Parameters | Works |
|---|---|---|
| ResNet-18 (He et al., 2016; Zhang et al., 2019a) | 11,689,512 | (Zhang et al., 2019a; Wang et al., 2020; Robey et al., 2022) |
| Wide-ResNet-8 (Zagoruyko & Komodakis, 2016) | 3,000,074 | (Shi et al., 2021) |
| CifarResNet-110 (He et al., 2016) | 1,730,474 | (Cohen et al., 2019) |
| CNN7 | 378,562 | (Shi et al., 2021) |
| Basic ConvNet | 1,663,370 | (Zhang et al., 2019a; Wang et al., 2020; Robey et al., 2022) |

Note that not all methods can be applied all model architectures. Table 7 summarizes the compatibility between the methods and model architectures. It should be noted that our method, along with ERM, DA, and PRL, applies to all architectures. We systematically evaluate each method for each model architecture to find the best-suited architecture for each method and dataset, *e.g.*, TRADES (Zhang et al., 2019a) eventually finds that ResNet-18 is the best matching architecture

---

**Algorithm 2** Inference With Certified Robustness

---

1: **Input:** Test data $\{\,(x_i, G_{x_i}) \mid i = 1, 2, \ldots, k\,\}$, given vicinity $B$, model (network) $h$, threshold $\kappa$, (statistical) significance level $\alpha$
2: **Initialization:** Certified Robustness *rate* $c \leftarrow 0$
3: **Output:** Model prediction $\{\,(x_i, \mathrm{pred}(x_i)) \mid i = 1, 2, \ldots, k\,\}$ for each case in test data; certified robustness for model $h$ on each case in test data against adversary at in vicinity $B$; certified robustness *rate* for model $h$.
4: **for** $i = 1, 2, \ldots, k$ **do**
5:     Predictions $S \leftarrow$ empty dictionary
6:     **repeat**
7:         Sample $t \sim \mathcal{U}(B(x_i))$ where $\mathcal{U}$ is a uniform distribution
8:         Prediction $s \leftarrow \arg\min_c \ell(h, t, c)$ where $\ell$ depends on loss choice
9:         **if** $s$ in $S$ **then**
10:             $S[s] \leftarrow S[s] + 1$
11:         **else**
12:             $S[s] \leftarrow 0$
13:         **end if**
14:         Most likely prediction $u = \arg\max S$
15:         Highest count $v = S[u]$
16:         Total count $w = \sum_j S[j]$
17:         $p_{\text{left}} \leftarrow$ Left-binomial-test$(v, w, p_0 = 1 - \kappa)$
18:         $p_{\text{right}} \leftarrow$ Right-binomial-test$(v, w, p_0 = 1 - \kappa)$
19:     **until** $p_{\text{left}} < \alpha$ or $p_{\text{right}} < \alpha$
20:     **if** $p_{\text{left}} < \alpha$ **then**
21:         Inference on $x_i$ is without certified robustness
22:     **else**
23:         Inference on $x_i$ has certified robustness
24:         $c \leftarrow c + 1/k$
25:     **end if**
26:     Prediction $\mathrm{pred}(x_i) \leftarrow u$ for $x_i$
27: **end for**

---

Table 6: Details on image classification datasets, and perturbation bounds for each task

| Task | MNIST | SVHN | CIFAR-10 | CIFAR-100 |
|---|---|---|---|---|
| Training Images | 48,000 | 58,606 | 40,000 | 40,000 |
| Validation Images | 12,000 | 14,651 | 10,000 | 10,000 |
| Testing Images | 10,000 | 26,032 | 10,000 | 10,000 |
| Image size | $28 \times 28$ | | $32 \times 32$ | |
| Color Channels | 1 | | 3 | |
| Classes | | 10 | | 100 |
| $L^{\infty}$ bound | 0.1 or 0.3 | | 2/255 or 8/255 | |
| Translation | | | $\pm 0.3$ | |
| Rotation | | | $\pm 35°$ | |
| Scaling Factor | | | $\pm 0.3$ | |

for SVHN, and the basic ConvNet for MNIST, while being not compatible to CNN7 (refer to Table 7). The most suitable architecture for each approach on different tasks is as follows: 1) For the MNIST dataset, all approaches except IBP can utilize the basic ConvNet architecture, while IBP adopts CNN7. 2) For the SVHN or CIFAR-10/100 datasets, all approaches except IBP or RS can use ResNet-18, while IBP utilizes Wide-ResNet-8 and RS adopts CifarResNet-110. In the following, we report the experimental results according to the most suited architecture.

In our training, we use different optimization strategies for different benchmarks to obtain the best performance. For example, we use Adadelta optimizer (Zeiler, 2012) with a learning rate of 1.0 for 150 epochs to optimize Basic ConvNet on MNIST. For the other three tasks, we use the SGD optimizer with an initial learning rate of 0.01 and weight decay of 3.5e-3. The learning rate for SGD

Table 7: Compatibility between different methods and model architectures

| Approach | ResNet -18 | Wide-ResNet -8 | Cifar ResNet -110 | CNN7 | Basic ConvNet |
|---|---|---|---|---|---|
| ERM | ✓ | ✓ | ✓ | ✓ | ✓ |
| DA | ✓ | ✓ | ✓ | ✓ | ✓ |
| PGDT | ✓ | ✓ | × | × | ✓ |
| TRADES | ✓ | ✓ | × | × | ✓ |
| MART | ✓ | ✓ | × | × | ✓ |
| RS | × | × | ✓ | × | ✓ |
| IBP | × | ✓ | × | ✓ | ✓ |
| PRL | ✓ | ✓ | ✓ | ✓ | ✓ |
| Ours | ✓ | ✓ | ✓ | ✓ | ✓ |

The Basic ConvNet architecture is suitable exclusively for the MNIST dataset, while tasks involving SVHN and CIFAR datasets require the utilization of residual networks with more parameters.

Table 8: Effectiveness metrics. To evaluate a model $h$ on any input data from some distribution $\mathcal{D}$, we assume that a test set $\mathcal{S}$ generalises the distribution, and $|\mathcal{S}|$ is the number of testing samples.

| Metric | Formula | Meaning |
|---|---|---|
| Standard Accuracy | $\frac{1}{|\mathcal{S}|} \sum_{(x, G_x) \in \mathcal{S}} \left( \mathbb{I} \left( \arg\max h(x) = G_x \right) \right)$ | The probability that the model's prediction is correct for an input from the data distribution $\mathcal{D}$. |
| Certified Robustness Rate | $\frac{1}{|\mathcal{S}|} \sum_{(x, \ ) \in \mathcal{S}} \left( \mathbb{I} \left( h(x) \text{ is with certified robustness} \right) \right)$ | The probability that the model's prediction has certified robustness for an input from the data distribution $\mathcal{D}$. |
| Certified Robust Accuracy | $\frac{1}{|\mathcal{S}|} \sum_{(x, G_x) \in \mathcal{S}} \left( \mathbb{I} \left( h(x) \text{ is with certified robustness} \right) \times \mathbb{I} \left( \arg\max h(x) = G_x \right) \right)$ | The probability that the model's prediction has certified robustness and this prediction is correct, for an input from the data distribution $\mathcal{D}$. |
| Defence Success Rate | $\frac{1}{|\mathcal{S}|} \sum_{(x, G_x) \in \mathcal{S}} \left( \mathbb{I} \left( \arg\max h(A(h, x, G_x)) = G_x \right) \right)$ | The probability that the model's prediction is correct when the input has been perturbed by adversarial attack $A$, for an input from the data distribution $\mathcal{D}$. |

$\mathbb{I}(\phi)$ a function that returns 1 if $\phi$ is satisfied and 0 otherwise

is reduced by a factor of 10 at epochs 55, 75, and 90. Our experiments are conducted on a server with an x86_64 CPU featuring 8 cores running at 3.22GHz, 54.93GB of RAM, and an NVIDIA RTX 2080Ti GPU with 11.3 GB of memory.

## C.2 Defending Against Adversarial Attacks

We evaluate our proposed method under 25 adversarial attacks on CIFAR-10 and compare its defence success rates with several baseline methods, as shown in Table 9. Row No Attack is the standard accuracy on the original testing set. It is evident that our approach outperforms all baseline methods across all adversarial attack algorithms. Except for the Pixle (Pomponi et al., 2022) attack, our method achieves a defence success rate of over 88% for all other attack methods. This is because Pixle attack focuses on searching for adversarial examples using the $L^0$-norm, which is not the focus of our method. Moreover, baseline methods with better average defence success rates, *i.e.*, PGDT, TRADES, and MART, exhibit a significant decrease in standard accuracy (more than 10%). PRL continues to show poor performance against these adversarial attacks, achieving a success rate of less than 5% in most of the cases (17/25). This is because the adversarial examples of a PRL model,

although only account for a small number (less than 9.38% on CIFAR-10, SVHN, or MNIST), are relatively easy to be searched by attack algorithms.

Table 9: Model defence success rates against adversarial attacks on standard benchmarks. Experiments are run on CIFAR-10 for all baselines and attacks.

| Attack | ERM | DA | PGDT | TRADES | MART | RS | IBP | PRL | Ours |
|---|---|---|---|---|---|---|---|---|---|
| No Attack | **94.85** | 94.21 | 84.38 | 80.42 | 81.54 | 89.45 | 48.40 | 93.82 | 94.23 |
| TIFGSM (Dong et al., 2019) | 35.10 | 33.00 | 65.70 | 62.90 | 69.10 | 45.40 | 40.20 | 34.00 | **92.80** |
| MIFGSM (Dong et al., 2018) | 0.00 | 0.00 | 50.90 | 51.90 | 50.50 | 5.80 | 38.10 | 0.00 | **92.80** |
| DIFGSM (Xie et al., 2019) | 1.00 | 0.00 | 51.75 | 50.50 | 53.60 | 4.10 | 38.10 | 3.10 | **92.80** |
| VMIFGSM (Wang & He, 2021) | 0.00 | 0.00 | 51.10 | 50.90 | 51.90 | 4.10 | 38.10 | 0.00 | **93.90** |
| TPGD | 38.10 | 39.20 | 69.30 | 69.10 | 70.10 | 48.50 | 50.00 | 28.90 | **91.80** |
| FGSM (Goodfellow et al., 2015) | 29.90 | 25.80 | 57.95 | 54.60 | 61.90 | 28.90 | 38.10 | 25.80 | **93.80** |
| RFGSM (Tramèr et al., 2018) | 0.00 | 0.00 | 49.15 | 50.40 | 48.50 | 3.70 | 38.10 | 0.00 | **90.00** |
| BIM (Kurakin et al., 2017a) | 0.00 | 0.00 | 52.00 | 57.20 | 47.40 | 2.10 | 38.10 | 0.00 | **90.70** |
| FAB (Croce & Hein, 2020a) | 1.00 | 2.10 | 43.00 | 46.40 | 40.20 | 5.30 | 38.10 | 4.10 | **90.10** |
| CW (Carlini & Wagner, 2017) | 0.00 | 0.00 | 32.20 | 35.10 | 29.90 | 1.00 | 38.10 | 1.00 | **92.90** |
| UPGD | 0.00 | 0.00 | 49.85 | 50.50 | 49.80 | 5.10 | 38.10 | 0.00 | **93.80** |
| FFGSM (Wong et al., 2020) | 19.60 | 23.70 | 60.55 | 55.70 | 66.00 | 33.00 | 42.30 | 29.90 | **92.80** |
| Jitter (Schwinn et al., 2023) | 11.30 | 12.40 | 48.15 | 47.40 | 49.50 | 34.00 | 39.20 | 24.70 | **90.70** |
| PGD | 0.00 | 0.00 | 57.40 | 54.60 | 60.80 | 7.20 | 40.20 | 0.00 | **91.80** |
| EOTPGD (Liu et al., 2019b) | 0.00 | 0.00 | 50.10 | 50.30 | 50.50 | 3.00 | 38.10 | 0.00 | **90.70** |
| APGD (Croce & Hein, 2020b) | 0.00 | 0.00 | 48.40 | 51.00 | 46.40 | 1.00 | 38.10 | 0.00 | **90.70** |
| NIFGSM (Lin et al., 2020) | 0.00 | 0.00 | 57.95 | 56.70 | 59.80 | 7.20 | 38.10 | 1.00 | **92.80** |
| SiniFGSM (Lin et al., 2020) | 4.10 | 1.00 | 59.00 | 56.70 | 61.90 | 23.70 | 38.10 | 12.40 | **93.70** |
| VNIFGSM (Wang & He, 2021) | 0.00 | 0.00 | 50.45 | 53.00 | 48.50 | 5.10 | 38.10 | 0.00 | **92.90** |
| APGDT (Croce & Hein, 2020b) | 0.00 | 0.00 | 40.90 | 44.30 | 38.10 | 0.00 | 38.10 | 0.00 | **88.70** |
| Square (Andriushchenko et al., 2020) | 0.00 | 1.00 | 50.40 | 54.00 | 47.40 | 3.10 | 38.10 | 2.10 | **88.08** |
| Add Gaussian Noise | 25.80 | 43.30 | 79.10 | 78.40 | 80.40 | 74.20 | 42.30 | 45.40 | **87.60** |
| OnePixel (Su et al., 2019) | 79.40 | 83.50 | 78.05 | 74.20 | 82.50 | 83.50 | 42.50 | 80.40 | **89.70** |
| Pixle (Pomponi et al., 2022) | 0.00 | 0.00 | 12.55 | 11.30 | 14.40 | 1.00 | 10.30 | 0.00 | **17.50** |
| PGDL2 | 1.00 | 0.00 | 35.80 | 36.10 | 36.10 | 5.20 | 36.10 | 0.00 | **92.90** |

$L^\infty$ bound at 8/255. $L^2$ bound at 10/255. For Gaussian noise, std=0.1. More detailed parameter setting is according to
https://adversarial-attacks-pytorch.readthedocs.io/en/latest/index.html

## C.3 IMPACTS OF HYPERPARAMETERS

In the following, we present the experiment results for adjusting hyperparameters other than $\lambda$.

**Vicinity size $\epsilon$.** To investigate the impact of the vicinity size on certified robust accuracy, we evaluate the models with altered $L^\infty$-norm radius $\epsilon$ on each dataset. Specifically, for MNIST, values of $\epsilon$ are selected from $\{0.1, 0.3\}$, while for the other three datasets, its values are chosen from $\{2/255, 8/255\}$. The results are shown in Table 10. We observe a trade-off between certified robust accuracy and the usefulness of certification, *i.e.*, decreasing the vicinity radius increases certified robust accuracy. Our approach achieves high certified robust accuracy ($> 85\%$) within a reasonable range of the vicinity and experiences a 0.36% increase with a one-third reduction and a 2.98% average increase with a one-quarter reduction.

**Percentage to certify $\kappa$.** To investigate how the strictness of certification requirement influences the certified robust accuracy, we vary the acceptable level $\kappa$ and significance level $\alpha$. The certified robust accuracy with regard to different acceptable level and significance level is presented in Table 11 and Table 12, respectively. Note that $\kappa = 0$ means conducting deterministic robustness certification on the model, which can only be achieved by IBP. The remaining baselines and our method can only provide probabilistic robustness certification results for the model. It can be observed that the variation of both the acceptable level $\kappa$ and significance level $\alpha$ does not have a significant impact on the certified robust accuracy, except ERM and DA. Specifically, for our method, when $\kappa$ has changed from $10^{-3}$ to $10^{-1}$, the certified robust accuracy has only improved by 1.05%; no increase in certified robust accuracy is observed when $\alpha$ varies from $10^{-3}$ to $5 \times 10^{-2}$.

Table 10: The certified robust accuracy of different approaches on various datasets within smaller vicinity.

| Approach | Certified Robust Accuracy | | | |
|---|---|---|---|---|
| | CIFAR-100 | CIFAR-10 | SVHN | MNIST |
| ERM | 33.45 | 48.85 | 59.34 | 48.01 |
| DA | 54.43 | 83.50 | 84.79 | 81.23 |
| PGDT | 44.59 | 83.23 | 87.98 | 95.89 |
| TRADES | 58.86 | 80.57 | 82.45 | 95.39 |
| MART | 56.73 | 81.35 | 73.84 | 95.22 |
| RS | 53.93 | 88.98 | 86.03 | 90.48 |
| IBP | 33.45 | 54.41 | 67.34 | 97.74 |
| PRL | 53.99 | 91.74 | 91.97 | 98.99 |
| Ours | **57.27** | **93.58** | **92.85** | **97.15** |

$\kappa = 10^{-2}, 1 - \alpha = 0.99$; $L^\infty$ bound at 0.1 for MNIST, and 2/255 for CIFAR and SVHN. See Table 1 for $L^\infty$ bound at 0.3 for MNIST, and 8/255 for CIFAR and SVHN.

Table 11: Comparison of the influence of different $\kappa$ values on the certified robust accuracy of CIFAR-10.

| Approach | $\kappa = 0$ (Deterministic) | $\kappa = 10^{-3}$ | $\kappa = 10^{-2}$ | $\kappa = 10^{-1}$ |
|---|---|---|---|---|
| ERM | - | 1.25 | 1.25 | 25.09 |
| DA | - | 73.50 | 76.07 | 86.59 |
| PGDT | - | 82.82 | 82.90 | 82.95 |
| TRADES | - | 78.69 | 78.80 | 79.60 |
| MART | - | 71.42 | 72.21 | 73.43 |
| RS | - | 87.63 | 87.98 | 88.08 |
| IBP | 35.13 | 39.98 | 40.00 | 44.41 |
| PRL | - | 89.88 | 90.63 | 91.97 |
| Ours | - | 91.73 | 91.75 | 92.78 |

For $\kappa > 0$, $\alpha$ takes $10^{-2}$.

Table 12: Comparison of the influence of different $\alpha$ values on the certified robust accuracy of CIFAR-10 where $\kappa = 10^{-2}$.

| Approach | $1 - \alpha = 0.95$ | $1 - \alpha = 0.99$ | $1 - \alpha = 0.999$ |
|---|---|---|---|
| ERM | 2.55 | 1.25 | 1.25 |
| DA | 77.56 | 76.07 | 76.07 |
| PGDT | 82.90 | 82.90 | 82.90 |
| TRADES | 78.80 | 78.80 | 78.80 |
| MART | 72.21 | 72.21 | 72.21 |
| RS | 87.98 | 87.98 | 87.98 |
| IBP | 40.00 | 40.00 | 40.00 |
| PRL | 90.63 | 90.63 | 90.63 |
| Ours | 91.75 | 91.75 | 91.75 |

