# OpenReview forum: "Towards Certified Probabilistic Robustness with High Accuracy"
_ICLR.cc/2024/Conference — ICLR 2024 Conference Withdrawn Submission_

### Official Review · Reviewer_d8Hi · 2023-10-17

**Soundness:** 1 poor
**Presentation:** 2 fair
**Contribution:** 1 poor
**Rating:** 3
**Confidence:** 3

**Summary:**

This paper presents an approach to training and inference that aims to achieve certified probabilistic robustness and still maintain high accuracy. The training method introduces a focus on minimising variance of the losses in a neighbourhood. Experiments are present to support the claims made.

**Strengths:**

There are a few main concerns with the paper (see Questions below) that need to be addressed. They may be based on a misunderstanding of the approach, and so appropriate answers from the authors would lead me to raise my score.

**Weaknesses:**

My main concerns are raised in the Questions section.

But some minor details:
- in section 2 it was stated initially that h outputed a label. And yet later on in the same paragraph this was changed to logits. Precision/consistency is required
- writing G_x = arg max h(x) for example needs to make it very clear what the argument is they're searching over (vector index of logit output here)
- It is claimed in the introduction that randomised smoothing suffers from "significant accuracy loss". This is a broad statement to make as for example one can vary the variance of the added noise in order to vary the certified robust radius achieved (and also the impact on accuracy)
- In equation (2) it is not clear what the random variable (more importantly its distribution) is for the inner Probability
- near end of section 2 ... "solution expects efficiency" should be "solution exhibits efficiency"?
- in (1) it seems to be defining an adversarial example as a pair of points x1 and x2. Yet later on single points are referred to as adversarial examples. The definition (1) is a bit misleading

**Questions:**

- In (3), conventionally I would have thought it should be G_x not G_t in the loss. i.e. you are trying to maximise the loss for the target label at x as t varies in the disk around x (also it is not made clear in the max specification that t is the free variable here)
The same problem occurs in the discussion below equation (5). The text talks of computing the loss against the label of x and yet the expression in the bracket "(i.e. E ....)" has G_t used as the label

- The distribution of t is not clear (in order to evaluate the expectation of the loss mean and variance). Is it uniform in the disk around x?

- My main question is more fundamental. Firstly, it is not clear what classifier's output is being certified. Is it h or h^\dagger

(i) If h, then it is not sufficient to show that adversarial examples are sparse in the disk of radius epsilon around x. This was implied in the statement "to establish certified probabilistic robustness, we must show that there is a guaranteed upper bound on the probability of adversarial example".  But it doesn't seem to be sufficient. e.g. if there is one adversarial point x' in the disk around x and an attacker has some algorithm that allows them to find it, then the point x' is adversarial, regardless of the fact that such points are very rare.

(ii) If h^\dagger (which seems more likely what is meant), then the intent is to say that the majority class value is the classification at x, certified on the disk around x if the other classes are (statistically) sufficiently small. [Separate note - I believe there may a difference between equation (6) setting h as the expected logits value and then doing argmax for the class on that, and algorithm 2 which does the argmax at each sample point and then counts the occurrences of each class and takes the max. This is a point which is discussed in the certified robustness literature]
If the claim that the vast majority of the points being of say class 1 in the disk is evidence that the classifier h\dagger is always 1 on the disk, this seems wrong. If evaluating h^\dagger at another point in the disk, we know nothing about the values outside the current disk. Hence the value of h^\dagger at a point where say its disk has less than half its area in common with the disk around x is unknowable. Approaches to randomised smoothing are able to make assertions about the values across a disk (of a radius which depends on the logit differences at x), given their long-tailed noise distributions.

Possibly this is a misunderstanding of what exactly is being claimed as being the certified robust classifier and what that property preciesly means. I look forward to clarifications.

---

### Official Review · Reviewer_Dkh4 · 2023-10-28

**Soundness:** 2 fair
**Presentation:** 2 fair
**Contribution:** 2 fair
**Rating:** 5
**Confidence:** 4

**Summary:**

The authors investigate a mechanism for improving certification, subject to training time modifications that purport to enhance robustness.

**Strengths:**

The authors of this paper have clearly placed a significant amount of effort into ensuring they comprehensively surveyed the state of the literature for certification mechanisms (although I did find it a little odd that foundational references like Lecuyer et. al. on DP for RS were missing).

**Weaknesses:**

Broadly my concerns relate to - the nature of the experimental comparisons constructed (and some scepticism about the associated results); non-standard evaluation measures for certification (not looking at the relationship between certification sizes and accuracy); and a writing style that makes the nature of the contributions difficult to parse (and a framing that is potentially disguising the similarities between the developed works and prior techniques - see the point about P > 0.5 below).

I also wonder if, given that one of the stated contributions is that it doesn't introduce "strong noise" (what exactly is strong noise?) at training time, that perhaps the evaluation metrics don't properly capture the areas that the paper has identified as being where the authors had made contributions.

I apologise that the following weaknesses are broadly chronological, rather than highlighting points in terms of their importance:
- Abstract doesn't include any performance metrics, which is something I'd suggest as a way to better establish the validity of your work from the start.
- Notation is odd. Equation 1 for example - eqn 1 is only a correct definition of an adversarial example if the distance condition is established. But the distance condition isn't a necessary part of things.
- The framing of the problem is also odd. "Unlike deterministic robustness, probabilistic robustness allows a small number of exceptions within the vicinity of a sample to have different labels". Mechanisms like Lecuyer / Li / Cohen / Cullen (who use randomised smoothing) can even have the majority of points within the vicinity of a sample to have a different label, just as long as the there is a majority of classes in the region with the greatest concentration of the probability density function. More broadly, the language is at times quite difficult to parse - there's nothing technically wrong with the writing, but the way it has been written makes the content difficult to parse, and disguises the nature of the contributions.
- I'm well versed in this field and Figure 1's caption is completely bamboozling. As far as I understand it, this is completely interchangeable to Lecuyer / Li / Cohen / Cullen - in that if the system is a binary classifier, then the minimum condition for constructing a certification is that the class probability (or expectation) is greater than 0.5.
- I'm concerned about the nature of the comparisons. Table 1 considers $\ell_\infty$ norm perturbations - but Cohen's randomised smoothing for example (the RS row) is for guarding against $\ell_2$ norm perturbations.
- Also the reliance on certified accuracy alone is odd - the certification literature typically considers the relationship between the size of the achievable certifications and the accuracy. What is being done here is essentially just a slice of that, and I don't believe it is fully representative of the dynamics that one would expect to see.
- Figure 2 has no Y axis label.
- If I understand the style considerations of ICLR, having two items side by side isn't in the spirit of the style guide. I'm specifically referring to Table 4 and Figure 2 being aligned in one block on page 8.
- How do the metrics from Table 4 change with dataset size? I'm also more than a little sceptical that every measure of inference time is 2.7 x 10^x for varying X except for your technique. Moreover, if I'm reading your Table 4 correctly, you have a 27 second inference time for most techniques, and a 72 hour inference time for Randomised Smoothing. Based upon performing experiments with Cohen et.al's code (your RS technique) on a RTX 1080 I am absolutely confident in saying that there is absolutely no way that this is correct for MNIST.
- Relating to table 4, in text you state that IBP requires the least inference time - except that's not true at all, given that most techniques require 27 seconds at inference time. Moreover, based upon my own experience with IBP I also must express a great degree of scepticism about these performance metrics for IBP. These results also call into question the fairness of the comparisons - a point which is complimentary to a point I make further on in this weaknesses section regarding the transparency of the architectures chosen.
- The code github link doesn't point to an appropriate repository - either submit it using an anonymous code submission platform, put it in the supplementary materials, or just state that you're going to publish the code on acceptance.
- I'd suggest that Cifar-10 and Cifar-100 are semantically similar enough that I wouldn't necessarily count them as 2 distinct datasets for comparison. Ideally I'd like to see Imagenet or TinyImagenet used, although I can appreciate with the hardware used by the authors such a dataset may be beyond their technical capacity.
- Unless I have misremembered things, there wasn't a great deal of specificity regarding how the architectures of each technique were chosen. The authors state "we report the experimental results according to the most suited architecture" - but most suited by what measure? How was this chosen, and what architectures were actually used for each technique in Table 4? Table 5 summarises the architectural features, but not their applicability.
- There's quite a reliance on statements without specificity - things like "strong noise" or "the hyperparameter \lambda must be reasonable".
- Algorithm 1, line 12 - according to this there's no dependency on the index i, so $\sigma_i = \sigma_j$ by definition, which then begets the question of why sigma is even indexed?
- The actual mechanism for achieving certification is also remarkably opaque.

For a paper that talks about training time modifications to improve certification performance, it's also odd that there's no consideration of Salman or MACER, which have already established training time modifications for improved certification. Their exclusion is surprising.

**Questions:**

If the authors could respond to some of the points raised within the weaknesses, that would be appreciated. While I understand that that was a long list of points, one of the core ones would be why Cohen et. al (as RS) was being used in an $\ell_\infty$ context?

One additional question I have relates to the variance minimisation in Eqn 5 - the variance is over distances less than epsilon, but that condition doesn't seem to align with the content of Algorithm 1 at all. Could you clarify this? Could you also discuss how many training samples are required to estimate the variance accurately?

---

### Official Review · Reviewer_iJ6m · 2023-11-01

**Soundness:** 2 fair
**Presentation:** 1 poor
**Contribution:** 2 fair
**Rating:** 3
**Confidence:** 3

**Summary:**

This paper builds upon previous works on probabilistic robustness and proposes a new method for training probabilistic robust classifiers, together with an algorithm for certifying probabilistic robustness. It is proposed to train classifiers on a combination of expectation and variance of the loss function, over perturbation distributions around a given input point. Experimental results show improved performance in terms of probabilistic robustness and adversarial robustness on MNIST, SVHN, CIFAR-10 and CIFAR-100 compared to baselines.

**Strengths:**

- The objective of minimizing a combination of expectation and variance of loss over the neighborhood of a point is a sensible (and to my knowledge novel) approach to achieving probabilistic robustness.
- The empirical evaluations seem strong, with state-of-the-art certified probabilistic robustness. The results on adversarial robustness (Table 3) are particularly impressive, appearing to significantly outperform even methods tailored specifically to achieve adversarial (rather than probabilistic) robustness.

**Weaknesses:**

- The novelty of the paper, aside from the variance training objective, is unclear. In particular, the suggested inference method (selecting the majority decision among sampled points) seems to just be randomized smoothing. Further, it is unclear how the certification approach compares to similar methods (Baluta et al. 2021, Zhang et al. 2023). I would suggest that the authors add an explicit related work section to clarify these points (as well as contextualize the work more thoroughly).
- The experimental methodology may not be completely fair as the baselines are using different neural networks (Table 5). I’m not entirely sure why the methods could not be compared using the same neural network (see question below).
- The correctness of the certification algorithm (performing left and right tests after each new sample) is unclear to me; in particular there is no proof provided for Thm B.2.

(Baluta et al. 2021) "Scalable quantitative verification for deep neural networks." 2021 IEEE/ACM 43rd International Conference on Software Engineering (ICSE)

(Zhang et al. 2023) “Proa: A probabilistic robustness assessment against functional perturbations”. Joint European Conference on Machine Learning and Knowledge Discovery in Databases (pp. 154-170). Cham: Springer Nature Switzerland

**Questions:**

- Is the inference method (Eqn 6, and the paragraph below) not just randomized smoothing? I would also note that the descriptions of randomized smoothing in the paper are somewhat inaccurate - randomized smoothing is not a training method, but rather a transformation of an input classifier into a smoothed classifier.
- To clear up my confusion, if we applied the “ERM” approach (i.e. training without variance term), and then applied the majority prediction as in Eqn (6), would this be equivalent to the approach of (Cohen et al. 2019)? If not, how is it different?
- Algorithm 2 appears to certify the original classifier $h$. On the other hand, it is suggested that the classifier you actually use for prediction is $h^{\dagger}$, i.e. taking the majority prediction of $h$ among sampled points in the neighborhood. Is this correct, and if so, is the new classifier $h^{\dagger}$ also certified?
- Similarly, to clarify, does the method “Ours” in the experiments refer to the original classifier $h$, or the “smoothed” classifier $h^{\dagger}$?
- How does your certification method compare to (Baluta et al. 2021, Zhang et al. 2023)? Also, can you explain/add the proof for Thm B.2 please?
- I do not understand Table 7 regarding the compatibility of various methods with different NNs. For example, why is PGDT and RS not compatible with some NNs? PGDT simply involves taking gradient steps to find adversarial examples, and RS is just a post-hoc smoothing operation.

Minor comments:
- Alg 2 Line 12: Should this be S[s] <- 1?
- It seems in Eqn (6) this should be a max over classes, rather than an expectation.

---

### Official Review · Reviewer_kTWZ · 2023-11-04

**Soundness:** 1 poor
**Presentation:** 2 fair
**Contribution:** 3 good
**Rating:** 5
**Confidence:** 5

**Summary:**

To achieve a practical balance between accuracy, robustness, and inference speed, the paper proposes a new combination of model training, inference, and certification. The training involves minimizing the variance of model outputs within a local region. The inference involves sampling a set of inputs and certification leverages exact binomial tests. Experimental evaluation is performed on models for the popular MNIST, CIFAR10, SVHN, and CIFAR100 datasets.

**Strengths:**

* Training for deterministic robustness can cause over-regularization and yield practically useless models. Randomized smoothing offers guarantees while reducing the loss of accuracy but the resulting model has a large inference overhead. The submission aims to overcome these important limitations of prior work by achieving less inference overhead, probabilistic guarantees, and minimizing accuracy loss.

* The particular combination of the training, certification, and inference algorithms for balancing robustness, accuracy, and inference speed is novel.

**Weaknesses:**

* The comparison in Table 1 using the proposed robustness metric is misleading. The guarantees provided by deterministic methods are fundamentally different from probabilistic ones. The probabilistic guarantees for local robustness with $\kappa=0.01$ are quite weak for high-dimensional inputs (going from $\kappa=0.01$ to $0.0001$ can be hard). If the authors insist on making such a comparison, then I would suggest doing the following:
(i) Count the total number of images within each adversarial region and then sum it across all images in the test set. Report the fraction of images correctly classified by each model, or
(ii) Show how the robustness scores and inference speed for different values of $\kappa$ vary as $\kappa$ approaches 0.

* Main mathematical details are not rigorously presented: (i) Proposition B.1 and Theorem B.2 are entirely stated in English which makes it quite vague, it should be possible to state these results formally, (ii) it must be possible to define the metrics in Table 8 more formally compared to just using indicator functions on statements in English, and (iii) sequential sampling seems to be important for the efficiency of the proposed method but it is never described in detail and its impact on the formal guarantees is unclear.

**Questions:**

* "The inference part of our method forces an attacker to generate “robust” adversarial samples ", why not evaluate your method against robust adversarial examples (e.g., https://arxiv.org/abs/1707.07397 or https://arxiv.org/abs/2007.12133)?

* " state-of-the-art certified training can result in up to 70% accuracy drop on MNIST and 90% on CIFAR-10" While there is a significant drop of clean accuracy with certified training, these numbers do not represent the accuracies obtained by state-of-the-art certified training methods. On another note, if there is 90% less accuracy on CIFAR10 then the resulting classifier is worse than a classifier that predicts the same class for all inputs.

* There has been some recent work on certified training for geometric perturbations (https://arxiv.org/abs/2207.11177). The authors should compare against it for their evaluation on rotation, translation, etc.

* Can you handle the composition of different geometric perturbations?

* The guarantees obtained here assume a uniform distribution in the local neighborhood. How realistic is this given that adversarial examples are usually distributed in specific subregions and given that the Pixle attack works quite well against your model? Can the method work with other distributions?

* While IBP is a standard method for certified training, it is currently not state-of-the-art. It will be better to report the numbers achieved by the latest methods (e.g., https://arxiv.org/abs/2210.04871 or https://arxiv.org/abs/2305.04574)